# The *Misericórdias* as Social Economy Entities in Portugal and Spain

**Antonio José Macías Ruano [1,*], José Ramos Pires Manso [2], Jaime de Pablo Valenciano [3] and María Esther Marruecos Rumí [1]**

[1] Department of Law, University of Almería, 04120 Almería, Spain; mmr646@ual.es
[2] Department of Business and Economics, University of Beira Interior, 6200-001 Covilhã, Portugal; jose.pmanso@gmail.com
[3] Department of Business and Economics, University of Almería, 04120 Almería, Spain; jdepablo@ual.es
[*] Correspondence: ajmacias@ual.es

**Abstract:** *Las Santas Casas de Misericórdias* (The Holy Houses of Mercy) are institutions of Portuguese origin that emerged in the late fifteenth century and that, over time, have expanded beyond the territories of the Portuguese Empire, including to Spain, where various *Casas de Misericordia* were created in their image and with similar purposes to the original. The *Misericórdias* continue to be relevant and present throughout Portugal, in various decolonized countries of the former Portuguese Empire, and in other territories that have been influenced by Portuguese emigration, and have always played an important role in the social care of citizens. In Spain, the *Santas Casas de Misericordia* do not have the same long history, nor the same social relevance as their Portuguese counterparts. However, even today, there are some *Casas de Misericordia* in Spain that provide social care services, having adopted various legal structures such as foundations, associations, and public entities.

**Keywords:** *Misericórdias*; social economy; Portugal; Spain

## 1. Introduction

The Holy Houses of Mercy are institutions of social care of Portuguese origin that emerged at the end of the fifteenth century for the development of the so-called actions of Christian mercy, guided by a deep religious feeling. They spread throughout the Portuguese empire, both in the metropolis and in all the territories discovered or colonized by it, with some of them achieving great economic, institutional, and social relevance such as those in Lisbon, Porto, Guarda, or Bragança in Portugal; São Paulo or Rio de Janeiro in Brazil; Luanda in Angola; Macao in China; and Maputo in Mozambique.

The Portuguese model of the Holy Houses of Mercy has also spread to other countries through the emigration of their nationals, with the recent creation of the Paris House in 1994 and the Luxembourg House in 1996. At the international level, entities representing the *Misericordias* have been set up, such as the "*União Europeia de Misericórdias*" or the "International Confederation of Charities/Misericordias," so that this type of charity and Christian service, far from being exhausted, is being perpetuated over time.

However, this type of brotherhood has had manifestations in other territories governed by other metropolises, such as the Italian ones in the mid-thirteenth century, or the Spanish ones that emerged immediately after the Portuguese ones and had similar characteristics—although they would soon be fundamentally differentiated from these as a result of the Council of Trent in the mid-sixteenth century.

The evolution of the *Misericordias* in Portugal has not been linear; although they have suffered various setbacks, they have always played an important role in social care for local citizens. Their

presence and actions have been, and are, fundamental to the Portuguese social welfare system. The *Misericórdias* work on behalf of the Portuguese government in caring for the most vulnerable—with some government financing. During the Enlightenment, the government confiscated the *Misericórdias'* assets, using its hospital and social care network to carry out its work of protection through the officialization of its resources; later on, they were finally integrated into the Portuguese Social Security service provision system.

In fact, in the current Portuguese Constitution, there is an implicit recognition of the *Misericórdias* for their social work; they are considered entities that are part of and collaborate with the current Social Security system, having been included in the legal category of Private Institutions of Social Solidarity (IPSS).

The current Portuguese Law on the Bases of the Social Economy[1], L. 30/2013, determines which social economy entities will have access to institutional and tax-related promotion and encouragement measures, including the *Misericórdias*, which in Portugal enjoy the status of a social economy entity, as an IPSS, but also have a particularly prominent position in the law.

In Spain, on the other hand, the existence and survival of the Holy Houses of Mercy does not have the same historical path, nor the social relevance. This is not because of their work of attending to the needy, which has also been very significant, but because of the different management and ownership of this type of entity and the public assumption of the social care service.

The differences between the Spanish and Portuguese *Misericórdias* are a matter of their initial conception and the impetus given by the monarchy, but the Council of Trent and its resolutions meant that the Spanish *Misericordias* came to depend mainly on the Church, while the Portuguese *Misericórdias* continued to be established by the monarchy, which meant that, practically from the outset, the development of the two was totally different.

The weakening of the periods of imperial development of the Iberian States, the liberal economic and political processes that affected Europe, the change towards parliamentary monarchies, the new concept of social care for the poor as an obligation of the government—in short, the historical course of both countries influenced the development of this type of entity in their respective territories: staying strongly in Portugal, and weakening in Spain.

In Spain, there has been no political action to promote or protect this type of private social care institution. Initially, those of an ecclesiastical nature were recognized and maintained, but the disentailments and the publication of the various welfare laws, against the backdrop of the assumption of the public nature of welfare provision, have made the survival of this type of entity very difficult. However, there are still a number of *Casas de Misericordia* in Spain today that provide social care services, having adopted various legal formulas such as foundations, associations, and public entities, which, on the basis of the provisions of Article 3.1.2 of the Constitution, have been set up to provide social care. Article 5 of Law 5/2011 of 29 March, on the Social Economy, stated that those bodies that have opted for a legal formula or structure recognized by the Law, such as foundations and associations, must be considered as social economy entities, which enables them to access the measures that are adopted at an institutional and legislative level to promote this type of structure.

The goal of this paper is to compare the development of the *Misericórdias* in the history of these two countries of the Iberian Peninsula. With a method of historical and comparative prospection, we will determine the causes of the different role of these entities in the social care system, as well as the different treatment of them at the legislative and institutional level.

---

[1]   It should be noted that the legal concept of the "Social Economy" and its legislative development in Portugal and Spain are very similar, with the respective Portuguese and Spanish laws on the social economy being similar. The list of entities eligible for this qualification can be updated to enable them to participate in the promotion and protection measures that the public sector provides for the promotion of this type of entity, as well as a series of advantages and benefits of a fiscal nature, such as the social security contributions provided for these entities (Macías Ruano and Pires Manso 2019, *passim*), so the relevance of their legal recognition as social economy entities in both Portugal and Spain has special relevance in terms of legal protection in both countries.

## 2. Historical Background of the Brotherhoods of Attention to the Needy

The *Santas Casas da Misericórdia* did not exist in a vacuum, but rather were the result of an evolution in the religious treatment and care of the needy. From the Late Middle Ages and during the Early Modern era, when economic growth began to take hold in Europe, new religious movements emerged that went beyond the Carthusian or Benedictine orders[2], giving rise to new mendicant orders such as the Franciscans, the Augustinians, and the Dominicans, who were generally concerned with poverty, the path to achieving glory, and the care of the poor and needy, that is, the Works of Mercy. Their activity was based on the belief that it is the actions of men that make them worthy of being saved or condemned in eternal life, and thus, good works are a means to salvation. Faced with this proposed ecclesiastical poverty, for the laity, the means to obtain eternal life was not the renunciation of material goods but rather adherence and service to God through men, which required time, attention, and devotion (Amorim and Pinto 2018, p. 55).

At the Second Council of Lyon in 1274, the idea of Purgatory as a theological space took shape. It was understood to be a place where the souls of the deceased were retained to await their atonement and purification; they could be relieved of their suffering by the prayers and pious acts of the faithful living (Zambrano González 2014, p. 1073)—an idea that was subsequently ratified by the Councils of Florence in 1439, where it was elevated to a dogma of faith, and by that of Trent, between 1545 and 1563. This new theological reality had an enormous impact on the economic sphere, since many deceased believers allocated part of their estate to the salvation of their souls with the commission of Holy Masses and completion of Works of Mercy in order to purify their souls and attain Eternal Glory as soon as possible.

A similar group that existed during the Low Middle Ages in Europe was the guilds, whose purpose was to defend the interests of the members of the same trade or craft, and to function as devotional and mutual aid societies that performed charitable acts only among their members. This was in contrast to the religious brotherhoods that cared for those in need (Russell-Wood 1968, p. 12), without distinction or on the basis of a professional relationship, and which were devotional in character and dedicated to social and charitable works (Díaz Sampedro 2011, p. 198). The religious or devotional brotherhoods, as associations of the faithful who gathered to practice religious worship in all its forms (Dos Guimaraes Sá and Lopes 2008, p. 12), incorporated and gave prominence to laypeople; their essential nature resided in the spirit of fraternity—*confraternitas*: literally, "with-brothers." They appeared from the end of the 12th century (Russell-Wood 1968, p. 2), with the 14th and 15th centuries being when their numbers increased significantly (Herrera Mesa 2006, p. 102). The devotional brotherhoods gained great relevance, especially from the 16th century, and became a religious way of life that superseded the legal frameworks of the Church, at least until the Council of Trent (1545–1563), wherein economic and functional control was established for existing and future religious orders, with the exception of those founded under the auspices of royalty. These brotherhoods also had an evangelizing function in teaching about the communion of saints through the practice of devotion, charity, or penance (García de Cortazar 2012, pp. 376–77). In essence, this meant that the salvation of the soul is aided by the Works of Mercy carried out in life, or by those who survive them to ensure, by atonement of sins, rapid transit from Purgatory to Eternal Paradise. At that time, charity and the care of the poor were financed by dispositions set out in the testamentary wills and testaments of wealthy people who applied to the religious orders or to the lay brotherhoods, whose service had already acquired an ecclesiastical dimension and were considered to be servants of the evangelical cause, in order that these brotherhoods carry out Works of Mercy on their behalf (Amorim and Pinto 2018, p. 54). The other source of financing came from the donations of the pious during their lifetime, either as a form of penance for their sins or for completion of Works of Mercy on their behalf, which would count towards the salvation of their souls.

## 3. The *Santas Casas de Misericórdias* in Portugal (Origins and Growth)

---

[2] The Benedictine order in its reformed Cluniac (10th century) and Cistercian (11th century) movements.

　　　Originating in Portugal, the *Santas Casas de Misericórdias* (Holy Houses of Mercy) have been providing social services to particularly vulnerable groups for more than five centuries. They are firmly entrenched in Portugal and still maintain extraordinary social relevance in that country as well as in other territories worldwide, whether due to historical colonial roots with a particular city or to the influence of a high level of Portuguese emigration.

　　　Although the first known *Cofradía de Nuestra Señora de la Misericordia* (Brotherhood of Our Lady of Mercy) was founded in Florencia[3], this brotherhood did not influence the establishment of the *Santas Casas de Misericórdias* that were created from the 15th century as brotherhoods that sought the completion of all Christian Works of Mercy (Russell-Wood 1968, pp. 3–4, 14–15). The first, which remains the most important of the Houses, was founded in 1498 in Lisbon by Queen Doña Leonor de Lencastre[4], the Trinitarian friar Miguel Contreiras and a group of six lay brothers. From their inception, the *Misericórdias* have overcome many obstacles, demonstrating a tremendous capacity for surviving all kinds of vicissitudes (Sá and Lopes 2008, p. 6). These vicissitudes have been both prosperous and adverse, as have been the historical moments in which their actions and trajectories have been developed.

　　　Aside from the social care provided directly by the Church in convents or parishes, the first manifestation of secular social philanthropy in Portugal occurred around the 11th century in the so-called *albergarias* (hostels, inns, or lodges) located along pilgrimage routes, usually old Roman roads, in which pilgrims were provided with a bed and small amounts of food and water for a limited period of three days. In case of illness, pilgrims could remain and receive the best medical care available at the time, however imperfect by modern standards of care (Russell-Wood 1968, p. 8). These hostels became a more defined aspect of health care and ended up being called hospitals when a distinction was made between mere accommodation and the treatment of ailments and the care for the sick. The hospitals became dedicated exclusively to the care of the sick, as in the case of the creation of the Orphan Children's Hospital in Lisbon, prior to 1258 (Sá 1995, p. 222), and the Hospital of the Innocents of Santorem in 1321; or the hospitals for elderly disabled people and for repentant fallen women that were founded in Coimbra by Queen Elizabeth of Portugal in the early 14th century (Pero-Sanz 2011, pp. 121–22). An important milestone in health care in Portugal was the foundation of the Caldas Hospital, in 1485, by Queen Doña Leonor, which was staffed by medical personnel, surgeons, barbers (for bleeding), pharmacists, and nurses for 100 patients (Russell-Wood 1968, pp. 11–12). Likewise, the *gafarias* (leper houses), such as those in Santarem or Lisbon, founded in the 12th century, were other major health care structures. While both types of establishments had very defined purposes and functions, neither enjoyed a generic scope of action that encompassed the plurality of the established Works of Mercy.

　　　When the Portuguese *Misericórdias* were founded at the end of the 15th century, in the Portuguese kingdom, as in other regions of Europe, there were numerous brotherhoods, both in rural

---

3　　The Venerable Archconfraternity of Mercy in Florence was founded by Piero di Luca Borsi in 1244 with the aim of maintaining bunk beds to transport homeless patients to the hospital.

4　　Dona Leonor de Lencastre, princess and queen of Portugal, for her exemplary life and constant practice of charity was called "Princess Most Perfect." She sponsored the arts and religious culture, with translations of works of the time and the printing of new editions. She supported writers, musicians, goldsmiths, painters, sculptors, architects, and many others. She was a woman of her time, of the Renaissance, and always guided by a deep spirit of Christian charity, of which the creation, not only of the first House of Mercy in Lisbon in 1498, or that of Óbidos in 1511, but also of the coastal hospital, with which the Church of Nossa Senhora do Pópulo was built, and the development of the city with the same name, Caldas da Rainha, is an example. In addition, there were the Convent and the Church of Madre de Deus, the Convent of the Announcements, the Church of Nossa Senhora da Merceana, the Church of Santo Elói, in Porto, the Chapels of the Monastery of Batalha and the seven shopkeepers in the Convent of Santo Agostinho (Brito Rebelo n.d., p. 5). Her Christian spirit led her to join the tertiaries of the Franciscan order in 1503 (Blick and Gelfand 2011, p. 130), and "protected the Franciscan Observants, the 'Lóios', the Hieronymites, the hermits of the Serra d'Ossa, among many other religious and clerics, founded the monastery of the Mother of God of Xabregas, on the outskirts of Lisbon, given to the Poor Clares of the Coventry. ... And until the end he conserved his immense fortune, which allowed him to do good to the poor and to all these institutions" (Rodriguez, 217: 123).

and urban areas—although the latter were more scarce (Sá 1995, p. 230)—that organized various forms of charity around the care of the poor and needy. Lay people, often royalty or royal houses, also sponsored the establishment of new mendicant movements (Sá and Lopes 2008, pp. 19–20). The emergence of the printing press also promoted the dissemination and knowledge of devotional works, expanding the belief in the usefulness of the Works of Mercy, already included in the *Summa Theologiae* of Saint Thomas, Part II-IIae, Question 32[5] (written between 1265 and 1274). This work, when dealing with alms, describes these acts of mercy and also describes seven types of alms of a corporal nature and another seven of a spiritual nature, taken from the description of the final judgment contained in the Gospel of St. Matthew, chapter 25, verses 31 to 40, and which were subsequently set in the Sum of Canisus or the Catechism of Trent (1556)[6]. The care of the poor, the creation of hospitals for the needy, the reception of newborns or foundlings, the aiding of helpless women, and the care of poor prisoners when neither they nor their family could pay for their upkeep/sustenance were the first areas of social intervention of the *Misericórdias* (Abreu 2000, p. 396). Other Works of Mercy such as the payment of ransoms for those who were captive in Islamic territories and were too poor to pay themselves, as well as the accompaniment and burial of those sentenced to death (Martins Da Silva 2016, p. 207), were a result of the Royal Decree of August 15th, the Day of the Assumption of the Virgin, in 1498, at the foundation of the first *Misericórdia* in the Kingdom of Portugal, in Lisbon, in the guise of a brotherhood, in the Chapel of the Pieta, or of Terra Solta (Sá and Lopes 2008, p. 24) and under the invocation of the Virgin Mary of Mercy, whose position as a mother allows her to intercede for men before God, using her mantle to cover and protect those who pray before her. This image of protection was used by the Cistercian Order in the first third of the 14th century, but was especially disseminated by the new mendicant orders during the 15th century (Lucía Gómez-Chacón 2014, p. 2).

During the era of colonial expansion, the temporary transfer of the Portuguese Court to cities outside the capital due to the frequent epidemics that Lisbon suffered from due to its port status (Russell-Wood 1968, p. 6) meant that the *Misericórdia* model founded in the capital was extended to different cities where the Court travelled: similar Houses were established in Setúbal, Évora, and Montemor-o-Novo in 1499, and in Santarém and Coimbra in 1500 (Sá and Lopes 2008, p. 28). New *Misericórdias* were subsequently founded throughout the Portuguese territory, assuming an essential role in Portuguese society by becoming its principal welfare institution.

The creation of new *Misericórdias* throughout the Portuguese territory became a priority for the Court, as well as in the newly discovered territories such as the Atlantic Islands; or conquered lands such as those in North Africa. The work of the *Misericórdias* was always focused on a spiritual interest in the care of the poor, the needy, and destitute. This was in addition to praying for souls in pain who hope to leave Purgatory and conducting Holy Masses commissioned by the deceased for their salvation. Payment for these services became a common practice and was a major source of income for the brotherhoods, who guaranteed the effective management of alms and legacies left by the wealthy and were considered to be intermediaries between the donors, generally wealthy people seeking absolution of their sins, and the needy poor, who were recipients of the Works of Mercy (Amorim and Pinto 2018, p. 65).

However, the charitable intention of the *Misericórdias* transcended the spiritual and religious context, and was proposed as a state policy to reinforce the presence of the monarch among the population, spreading the idea of a closeness between rulers and those being governed, an idea that did not exist in reality (Abreu 2000, p. 397). It was also an instrument, not for the centralization of

---

[5]  http://hjg.com.ar/sumat/c/c32.html.

[6]  From the first Catholic Catechism of the Council of Trent, until the current one of 1992, the Works of Mercy have been defined as charitable actions through which the neighbor is helped in his bodily and spiritual needs, being those that attend to the first aspect, to the corporal, those of visiting and caring for the sick; to feed the hungry; give water to the thirsty; give shelter to the pilgrim; clothe the naked; visit the imprisoned; and bury the dead. With regards to the spiritual, these include: instructing the ignorant; correcting the one who is wrong; counseling those who need it; forgiving offenses; comforting the afflicted; bearing patiently the wrongs of others; and praying for the living and the dead.

charity, but for the autonomous provision of care and hospital services, with papal support being granted in 1499 through the "*Cum sit carissimus*" Bull, which authorized the unification of all small hospital centers into a single larger one in Coimbra, Évora, or, in the case of the *Misericórdias*, in Santarém, thus handing them over to local elites who would take over their management (Sá 2001, p. 340). The economic strengthening of these brotherhoods was also sought by granting them special privileges such as access to prisoners, the cleaning of prisons in the city of Lisbon, and the general care and support for poor prisoners, which involved costs, for the relief of which they were granted a monopoly on collecting alms and making collections (Russell-Wood 1968, pp. 17–18), as well as, from 1503, a percentage of the rents and contracts of the Royal Treasury allocated to charities by King Manuel I (Sá 2001, p. 340). Subsequently, as recorded in the Bull to the Mercy of Lisbon of 20 August 1545, the assignment of the testimonial legacies of the pious that were not fulfilled in the time indicated by the testator were granted to the *Misericórdias*. This was reinforced by the Law of November 1564, which extended this privilege and assigned any testamentary legacy of Lisbon that was not fulfilled in time to this Mercy.

These new Portuguese brotherhoods of *Misericórdia* were elitist entities that were formed by a *numerus clausus* of members (with equal number of noble brothers and *oficiais mecânicos*), in which initially both men and women[7] were accepted, governed by a *compromisso*, a Statute issued by the King. Having been founded by a Royal House and, because of the opposition of Portugal to the ecclesiastical intervention in the control of the brotherhoods, having been, since the Council of Trent, granted special status and a singular structure (Abreu 2000, pp. 398–99), by means of a statute of "immediate royal protection," meant they were widely favored by the Crown (Lobo de Araújo 2014, pp. 539–40). The fact that the Portuguese monarchy preserved the secular character of these brotherhoods was fundamental for their later development, contrary to what happened with the Spanish Mercies.

In the process of expansion of the *Santas Casas de Misericórdias* throughout the Portuguese territory, all Houses were associated with the one in Lisbon, which became a kind of Arch Confraternity with many associated Houses, making the capital city much more relevant than those established in the rest of the territory.

In this process of strengthening the network of the *Santas Casas de Misericórdias* throughout the territory, which was in the interest of the Portuguese monarchy, the transfer of the Royal Hospital of All Saints of Lisbon to the *Misericórdia* of Lisbon in June 1564 was particularly important as it then went on to take over all the sources of income, including donations, testamentary dispositions, and legacies, initiating a process of continual transfer of all the Portuguese hospitals to the *Misericordias* in each municipality, resulting in their eventual control of the national hospital. This is turn meant that the main activity of the *Misericórdias* was the care of the sick, although they did not abandon the other 13 Works of Mercy set by the Church (Abreu 2000, p. 401). Concurrent to the process of hospital monopolization in the hands of the *Misericórdias*, there was, by statute, a nobilization of the administration of these brotherhoods, with the roles of Clerk and Treasurer being assigned exclusively to nobles (Abreu 2000, p. 402), so that the control of the population in an area as sensitive as that of public and individual health throughout the Portuguese territory was controlled by the ruling class, who, by means of the endowment of the *Misericordias*, gained economic control at a local level that empowered them to such a degree that, by the beginning of the 17th century, the Portuguese *Misericórdias* were managed by the ruling class, who had control of public aid and a monopoly on the national hospital network, and became the recipients, via testamentary dispositions, of the estates of penitent souls that sought to be released from Purgatory via the payment of perpetual masses, all of which gave patrimonial consistency to these brotherhoods, enriched their managers, and defrayed hospital expenses, freeing up demands on the local coffers and public treasury (Abreu 2000, pp. 404–5).

---

[7]   From its inception, the brotherhoods accepted both men and women as members, until in 1577 the statutes of the Mercy of Lisbon were changed and women were excluded—although with some exceptions such as the *Misericórdia* of Montijo, where women had always been admitted as brothers. Women were once more generally allowed to become brothers in the late 19th century, being admitted again, in general, at the end of the 19th century (Lopes 2002, p. 92).

Power and money do not generally lead to honesty, and the power and wealth generated by the *Misericórdias* caused the neglect of their testimonial and posthumous commitments to such a degree that this breach of the redemptive commitments of the brotherhoods with regards to celebrating Perpetual Masses on behalf of the souls in Purgatory was made public and notorious. This led, from the beginning of the 18th century, to a significant reduction in the number of Perpetual Holy Masses commissioned for the deceased. The Portuguese Church itself condoned a large number of pending masses owed by the *Misericórdias*.

The *Misericórdias*, being governed by nobles, granted their members credit or loans with very low or no interest. The nobles, in turn, took advantage of this and did not always pay these loans back. This meant that, together with the reduction in the value of property rental incomes due to price inflation, and the reduction of commissioned Perpetual Masses in testamentary dispositions, the economic status of the Mercies became more and more precarious. As a result, the *Misericórdias* often had to resort to private loans for the provision of their services and, in the event that they were not able to pay back these loans, they were forced to sell their properties, resulting in even further impoverishment and discredit of the brotherhoods (Lopes 2002, p. 79).

This situation led to the enactment of various laws of a marked interventionist nature, sponsored by the Marqués de Pombal[8], such as the Law of 4 July 1768 against the sale of property of the *Santas Casas de Misericórdia* to its members, which was the first confiscation of property improperly owned by religious administrators; or the Law of 9 September 1769, with the establishment of testamentary limitations for the pious legacies willed to the *Misericórdia* of Lisbon, prohibiting the naming of the "soul" as the heir of the estate of the deceased—although later, given the scarcity of resources left for them, it was allowed, by the Law of 31 January 1775, that testators without relatives to the fourth degree could freely dispose of their property in favor of brotherhoods (Abreu 2000, p. 410).

The peculiar nature of the *Misericórdias* during this period corresponds to the different economic and financial crises that Portugal suffered during the second half of the 18th century. Thus, the decreased importation of gold from Brazil caused an acute economic crisis in the last third of the century, which was compounded by the French invasions of the early 19th century and the economic and commercial detachment of Brazil, its main colony, which signed an agreement with England in 1810, produced a significant disaster for the national economy and an impoverishment of its citizens, which led to the spread of epidemics and the inability, due to saturation, of care institutions to respond to the needs of citizens, (Lopes 2002, p. 80). These crises also led a large part of the population to emigrate, mainly to Brazil. Around 700,000 people left the country during the 18th century, creating a significant Portuguese diaspora (Pérez 2012, p. 4).

However, while in Europe Enlighted Despotism was concerned with consistent public health reform, the concern of the Portuguese Crown focused on the legacy of the *Misericórdias*, now with fewer resources and with the same oligarchic leaders, which, in fact, led to a general worsening of health care throughout the kingdom. So limited was the financing of the *Misericórdias* in the last quarter of the 18th century that the Office of Mercy and the Royal Hospitals of the Sick and Abandoned Children asked Queen Doña Maria I to grant a royal permit to establish an annual Lottery "to take advantage of the urgent needs of the so-called two hospitals" (Decree of 18 November 1783)[9]. This permission was granted to different *Misericórdias*, but only that of Lisbon was successful and it was the only one to survive the Decree of 23 September 1828, which prohibited all raffles and lotteries except those granted to the *Casa Misericórdia* and the *Casa Pía* in Lisbon, created in 1780 for the care of beggars and the education of orphans (Lopes 2002, p. 80).

The Decrees of May 1800 and 18 October 1806 for the restructuring of public aid, which continued to consider social care from a paternalistic, charitable, and pious perspective, failed in their

---

[8]　A historical figure of the mid-18th century in Portugal who became Prime Minister under King José I and went on to lead the greatest economic, social, and administrative reforms of Portuguese Enlightened Despotism, such as the materialization of legislative and administrative centralism, creating the Royal Bank, taking charge, with full powers, of the reconstruction of Lisbon after the Earthquake of 1755, and controlling the Court of the Inquisition, eliminating the *Autos de Fe*.

[9]　http://www.scml.pt/pt-PT/santa_casa/historia/#seculo_xviii.

attempt to force the *Misericórdias* to cede their properties and establishments to the Crown, in line with the confiscation processes that ensued in Europe. It was only due to the intervention of the so-called *Provedores* (Regional Presidents) that the country succeeded in bringing all the *Misericórdias* of the kingdom under the same regulations according to the *Compromisso* Statutes of the *Misericórdia* of Lisbon.

In the aftermath of the French Revolution and other European liberal movements against old regimes, the political situation in the early 19th century in Portugal was aggravated by the so-called Liberal Revolution of 1820 in Porto, which resulted in a change of government in which King Juan VI went from being an absolute monarch to a constitutional ruler, and the publication of the first Portuguese Constitution in 1822, which had intermittent and limited validity. The political situation worsened even further after the death of John VI with the so-called Liberal Wars, which lasted until 1834, with the second appointment of Pedro IV as the new, ephemeral King of Portugal. In the first phase of Portuguese liberalism, the ideas of the Enlightenment regarding the State's duty of care, not as a paternal figure but rather as a guarantor of the health and wellbeing of its citizens, took hold. Despite these new ideas, the economic reality at the time did not allow for the creation of a public social care system, with the State having to, instead, rely on the welfare structure that had taken root over many centuries in the form of the network of the *Casas de Misericórdia*. It was deemed necessary, however, to wrest control from local leaders (Lopes 2002, p. 86).

The economic situation of these brotherhoods at that time was very precarious, and their management poor and unhealthy. So serious was the situation that the *Misericórdia* of Lisbon was managed by an Administrative Commission appointed by the Duke of Bragança, Pedro IV, who was acting as regent for his daughter, Maria II, and without the intervention of any of his fellow members of the brotherhood of the *Misericórdia*. This Commission was not operational, and was therefore dissolved by Decree on 2 December 1851, with the *Misericórdia* of Lisbon going on to be administered by a royally appointed *Provedor* and four deputies, two of them appointed by the Brotherhood of *Misericórdia* (which never actually took place) and two others elected by the Government, which meant, in fact, the disappearance of the *Casa Santa* (Holy House) as a brotherhood of a private nature, transferring control to public administration.

The situation was exacerbated by the fact that, while the profits generated by the lottery diminished considerably, at the same time, the application of the Laws of 4 April 1861 and, particularly, the Law of 22 June 1866, regarding the confiscation of property not currently in use for pious and charitable works meant that the *Misericórdias* had to sell off some of their properties in order to use the income from the sales to purchase public debt. In principle, this investment in public debt was a relief for some *Misericórdias*, which were then able to receive some income from public credit and use their new found liquidity from the creation of indirect rural banks, an idea developed by the Andrade Corvo Law of 22 June 1867.

However, the national financial crisis at the end of the century and subsequent inflation proved fatal for the *Casas de Misericórdia* due to the Law of 26 February 1892, which reduced the interest rate on public debt by 30% (Lopes 2002, pp. 88–89), further exacerbating their financial situation. The situation was only mitigated by the reforms carried out during the appointment of the *Provedor* of the *Misericórdia* of Lisbon, the Marquis de Rio Maior (1870–1888), who lightened some of the burdens that the *Misericórdias*, including the so-called "*Instruções Regulamentares sobre o Serviço de Vigilância e Polícia da Roda*" (Regulatory Instruments for the Service of Surveillance and Control of Foundling Wheels"—founding wheels being instruments used for the reception of foundlings and the verification of the identity of their parents, which dramatically reduced the number of foster children—as well as the establishment of new subsidies covering the entire period of breastfeeding of the child and the granting of premiums to mothers who for the first year collected their children from the foundling home, with the subsequent saving for the *Misericórdias*. In 1887, they established the so-called "*Sopa da Caridade*" (Soup of Charity) as a preventive measure against diseases caused or aggravated by malnutrition. This measure came into effect in March 1888 and, by 1894, had become such a regular feature that it was known as "the first and main alms of the Holy House" (Forte Cordeiro 2012, p. 26).

The beginning of the 20th century saw a change in national policies that led to a substantial change in the operation and functions of the *Misericórdias*. With the Hintze Ribeiro Decree of 24 December 1901, national oversight agencies for health and charitable services were created. In 1903, when a new law on public aid was being planned that would signify greater public intervention of the *Misericórdias*, there was, at their request, the First Portuguese Welfare Congress, which was held in Porto in January 1905. The Congress concluded that there was a need for the *Misericórdias* as a principal institution of care and charity, and that they should be overseen and protected by the public administration. This led to more national congresses throughout the century, and the federation of all the Houses, in 1976, as the *"União das Misericórdias Portuguesas"* (Union of Portuguese *Misericórdias*), a milestone in the history of these brotherhoods (Lopes 2002, pp. 93–94).

Following the regicide of King Carlos I of Portugal and his son Luis Filipe in 1908, the son and little brother, respectively, of those killed, Manuel II, ascended to the throne; after various changes of government, within just two years he went into exile as a consequence of the establishment of the Portuguese Republic. This new political situation did not especially affect the *Misericórdias*, which continued to be recognized by the new State. However, Portugal's participation in the First World War and the Spanish Flu that devastated the nation between 1914 and 1918 proved fatal to the survival of these brotherhoods. The income the *Misericórdias* received, following the confiscation of the previous century, mainly came from interest-bearing investments in public debt, and the inflation brought about by Portugal´s entry into the war, which lasted until the 1920s, brought about a general impoverishment of the country and, by extension, the Holy Houses. Despite this, a great many *Misericórdias* were founded during the First Republic, thanks to the republican legislation of the time adopting the Constitution of 1911, which recognized the need and right to public aid. Thus, laws were published such as the one of 20 April 1911 regarding the Separation of Church and State, which limited how corporate funds could be spent on religious worship, with the consequent saving for the brotherhoods, which could now focus their resources solely on care.

On 25 May 1911, a law was passed that restructured health care and created organizations such as the National Public Health Council, which includes the *provedores* of the *Misericórdias* of Lisbon and Porto. In 1916, the *Misericórdias* were exempted from the payment of stamp duty, judicial, administrative, and fiscal costs; and in 1919 the National Institute of Compulsory Social Security and General Welfare was created, with one aim being the protection of charitable institutions (Lopes 2002, pp. 95–96). The above are examples of legislation that reinforced the social and welfare work of the *Misericórdias*.

In 1924 the First Congress of *Misericórdias* was held in Elvas, with the participation of 306 Houses from across the country, and the presence of the Government, which resulted in a turning point for the economic bailout of these brotherhoods and their empowerment. Laws were immediately enacted such as the one of 29 June 1924 that resolved a large part of the shortfalls of the *Misericórdias* that provided social care, or the Law of 8 September 1924, which meant that the *Misericórdias* would now receive funding directly from the State. Finally, on 1 November 1924, by means of Law Decree 10242, the Government recognized that the health care provided by the *Misericordias* in each Council was mandatory, and thus new legislation was enacted to promote the creation and maintenance of numerous brotherhoods during the First Portuguese Republic (1910–1926) (Lopes 2002, p. 99), expressly recognizing the *Misericordia* of Lisbon as an official public health institution (Article 12)—a label that, in fact, it already possessed, since its public intervention in 1851.

After the military coup d'état of 1926, a National Dictatorship was installed that lasted from 1926 to 1974. Known as the New State from 1932 and under the auspices of the dictator Oliveira Salazar, public health was merely symbolic compared to that developed by the *Misericórdias*. As reflected in the Decree of 23 July 1928, the *Misericórdias* were recognized as pivotal organizations, acting at the local municipal level in an advisory and coordinating capacity. In the Second Congress of the *Misericórdias* of 1929, the issue was raised as to whether they should be considered a religious organization, even though they had never been supervised or governed by an ecclesiastical hierarchy, but rather had always been administered by nobles and lay people performing pious Works of Mercy. In the Administrative Code of 1940, the merits were attributed to the legal nature of canonically

established associations for the practice of Christian charity, which could lead to a conflict over the ecclesiastical control of the same. However, the Decree Law of 7 November 1945 specifies its civil nature in that its statutes—*compromissos*—had to be approved by the Ministry of the Interior. This notwithstanding, the intervention of the ecclesiastical hierarchy in the *Misericórdias* became more and more evident in its management, with the result that some *Misericórdias* decided to turn away members who were not practicing Catholics.

During the 1950s and 1960s, new *Misericórdias* were established, sponsored by legislation that permitted them to acquire new properties and by the increased income from the newly created Totobola, a Spanish betting game. However, since the early 1970s there has been a conceptual change in public health care with the publication of Decree Law 413/71 and Decree 351/72, which sought to grant hospital control to public administrations, with provisions made for the nationalization of hospitals' care services as a result of the Carnation Revolution of 25 April 1974 (Lopes 2002, pp. 101–6).

In April 1974, the Carnation Revolution, a military uprising against the Salazar dictatorship, established democracy in Portugal. Under this new political regime of a marked socialist character, banking and a large part of the industrial sector were nationalized. As far as the health care sector was concerned, the nationalization of hospitals was agreed by means of Decree Law 704/74 of 7 December, which established that the hospitals would be managed/administered by government commissions, albeit while retaining private ownership of the properties. The Constitution of the Republic of Portugal of 2 April 1976, initially espousing socialist principles, was, according to Article 2, intended as a "transition to socialism." However, in regulating social rights and duties, in its original Article 63.3, it supported the existence of so-called private social solidarity institutions—currently in Article 63.5—that provide health and social care outside the public framework. The new political situation and legislation led to the celebration of the 5th Congress of *Misericordias* in November 1976, and the emergence of the *União das Misericórdias Portuguesas* (Union of Portuguese *Misericórdias*) (UMP), which would become a body recognized by the State in what would later be known as the Statute for Private Institutions of Social Solidarity (Almeida 2010, p. 159).

Decree Law 519-G2/79 of 29 December 1979 regulates the Statute of Private Institutions of Social Solidarity (known in Portuguese by the acronym IPSS) and in Article 3, included the *Hermandades de Misericórdia* (Brotherhood of Mercy) among cooperatives and solidarity foundations, recognizing their legal identity as a public utility: "The State exercises in relation to the institutions guiding and guardian action, which aims to promote the compatibility of its purposes and activities with those of the social security system, ensure compliance with the law and defend the interests of the beneficiaries and the institutions themselves"(Article 6). Thus, the *Misericórdias* were uniquely regulated in the law itself, in Articles 56 to 61, which, in addition to granting legal recognition to them as a canonical institution, linked them to public coordination and order concerning the development of social care activities and recognized the uniqueness of the *Casa Santa de Misericórdia* in Lisbon, defined as a Public Institute that is governed by special legislation (Article 61). In addition, by the Resolution of the Council of Ministers of 2 February 1980, the improper nationalization of the *Misericordias* resulting from the application of Decree Law 704/74 of 7 December 1974 and the need for compensation were also acknowledged. This came into effect with Decree Law 14/80 of 26 February. The above notwithstanding, the ecclesiastical nature of the *Misericordias* remained, although they were considered private associations of the faithful. The Portuguese Episcopal Conference of March 1988 published the General Norms for the Regulation of Faithful Associations pertaining to the new Canon Law of 1983, which normalized the legal status of some *Misericordias*, such as the *Misericordia* of Pará, Brazil, founded in 1667, which had originally been founded with ecclesiastical authorization and only later requested royal confirmation (Lobo de Araújo and Paiva 2007, p. 9). There is, however, an ongoing argument since the late 1980s about whether the *Misericórdias* should be considered lay or ecclesiastical entities. This conflict has involved a series of incidents and disagreements that have ended up in the courts of the Holy See (Lopes 2002, pp. 107–10), which are still discussing their status and management.

## 4. The *Misericórdias* as Social Economy Entities in Portugal

As previously stated, the Portuguese Constitution of 1976, in its initial format, clearly espoused socialist principles: "The economic–social organization of the Portuguese Republic is based on the development of socialist production relations, through the collective appropriation of the main means of production and soils, as well as natural resources and the exercise of democratic power of the working classes" (Article 80). Public interventionism was the dominant message. However, with the successive modifications to the Constitution in 1982 and in 1989, the precepts dedicated to the economic organization took on neoliberal tendencies. Thus, today, Article 80 is dispossessed of any mention of socialist indoctrination, assuming a mixed production system, including the fundamental principles of social economic organization, among which were the "Coexistence of the public, private sector and the cooperative and social sector of social ownership of the means of production." Furthermore, in Article 82, the three sectors—public, private, and cooperative/social—are defined, specifying in Section 4 that the third sector is made up of "a) The means of production owned and managed by cooperatives, subject to cooperative principles, notwithstanding the specialties established by law for cooperatives with public participation, justified by their special nature; b) Community means of production, owned and managed by local communities; c) The means of production subject to collective exploitation by workers; and d) The means of production owned and managed by non-profit legal persons, whose main objective is social solidarity, especially entities of a mutualistic nature."

In general, the constitutional recognition of the private sector and, within it, the social economy, both cooperative and social, stands out, although demanding that a not-for-profit purpose be pursued for the achievement of social solidarity. The necessary condition for any organization to be included within the social economy is to have an economic activity and be non-profit, such as: (1) The entire cooperative and social sector, as set out in the Portuguese Constitution; (2) Other foundations that have an economic activity; (3) Associations with a social purpose that also have an economic activity; and (4) Commercial companies whose social interests correspond to entities integrated in the cooperative and social sector or, which belong to one of the above (Namorado 2006, pp. 15–19).

If we focus on the singular normative scope of the entities that are part of the cooperative and social sector, as far as associations and foundations are concerned, these are substantially regulated in the Civil Code of 1966; and in relation to cooperatives in its Cooperative Code, initially set by Law No. 51/96 of 7 September, repeatedly amended, and currently in present Law No. 119/2015 of 31 August.

Another singular legal formula that has constitutional recognition in social care is that which was included in the initial Article 63.3 of the Portuguese Constitution recognizing the so-called Nonprofit Private Institutions of Social Solidarity (IPSS), which, following the amendment of the constitutional text of 1982, changed the qualification of private to that of private individuals. With the reforms of 1989 and 1992 of the constitutional text (partial reforms), Section 3 of Article 63 was changed and Section 5 was incorporated; due to the reform instituted by Constitutional Law 1/1997 with the current text of Article 63.5, the support and necessary supervision of the State is recognized "in the activity and operation of private social solidarity institutions and others of recognized interest from the non-profit public," within the framework of the Social Security system.

The constitutional recognition of the IPSS, since its original draft in the Constitution of 1976, and after a consultation between the State, the Union of *Misericórdias* and the Episcopal Conference (Branco 2017, p. 544; Amorin 2018, p. 35 ), there was some legislative development of these private entities, which occurred with Decree Law 519-G2/79 of 2 December, which approved its Legal Statute, although, as explicitly stated in its preamble, there are regulations specifically dealing with these entities such as the Administrative Code of 1940, Law No. 1998 of 15 May 1944, or Law No. 2120 of 19 July 1963. Within this regulatory framework a distinction was made between IPSS of an associative nature—social solidarity associations of volunteers and *Misericórdias*—and of a foundational nature—social solidarity foundations—(Briones Peñalver et al. 2012, pp. 41–42), and thus the IPSS universe grew, with the establishment, in 1980, of the Union of Private Institutions of Social Solidarity (UIPSS), renamed in 2008 as the National Confederation of Solidarity Institutions (CNIS) (Branco 2017, p. 545).

As stated in Decree Law 519/79, IPSS Statute, within this category, which is not strictly an organizing form but a legal statute granted to entities that are the organizational expression of solidarity born from civil society (Ampudia de Haro 2017, p. 154), the Brotherhoods of Mercy are grouped together with associations, foundations or cooperatives of social solidarity, voluntary associations of social action, and mutual relief associations (Article 3). That is, the Brotherhoods of the Holy Houses of Mercy, with the exception of that of Lisbon, which has been a Public Institute since 1851 and governed by special legislation (Article 61), are among the structures of the IPSS. The specific regime is contained in Articles 56 to 60 of the Statute, and they are institutionally represented by both the UMP and UIPSS, with the *Misericórdias*, therefore, having a role of special relevance in in all social and welfare policy after constitutional recognition and the legislative development of IPSS.

The IPSS parameter includes a heterogeneity of legal structures, yet they are treated in all areas as a homogeneous group (Carneito 2006, pp. 235–36); as such, all organizations included under the IPSS Statute automatically acquire the status of public utility legal entities that enjoy various exemptions and tax advantages (Almeida 2010, p. 160).

As part of the IPSS, the *Santas Casas de Misericórdia*, are entities of public utility. That is, the legal nature of these institutions, regardless of their canonical nature, has the legal status of a public utility of a private nature, with full capacity to act, and they are framed within the Social Security system, under the protection, guardianship, and guidance of the State, which coordinates and subsidizes them—Arts. 2 and 4 of Decree-Law 519/79.

The Statute for the IPSS of 1979 has been modified at different times. According to its latest modification and current draft, Decree Law 119/83 of 25 February and Decree Law 172-A/2014, the *Santas Casas de Misericórdia* form part of the IPSS (Article 2), with a legal framework stated in Chapter 2 of the governing law (Articles 68 and 71) specifically recognizing their identity as an association established in accordance with canon law, with the objective of providing social services and practicing acts of Catholic worship based on Christian morals and doctrine and thus being a supplementary regime to social solidarity associations.

The IPSS were framed, initially, within the scope of Social Rights and Duties, in Social Security and Solidarity (Article 63 CP); after the institutionalization of its Statute—Decree-Law 519/79—the collaboration procedures were also regulated by the State through the so-called Collaboration Protocols (Ampudia de Haro 2017, p. 153). In 1984 Law No. 27/84 of 14 August on Social Security was published, with an initial provision—Article 1—stating that the law "defines the foundations of the Social Security system provided for in the Constitution and the social action carried out by social security institutions, as well as private non-profit initiatives for similar purposes to those institutions." This was followed by a legal framework for IPSS in the field of participation in Social Security (Article 61) and its relationship with the State (Articles 66 and 67). The Social Security Act of 1984 was repealed by Law No. 17/2000 of 8 August, which continued to consider IPSS to be social security collaborative entities (Articles 37, pp. 101–33). In addition, Law 32/2002 of 20 December, which repealed Law 17/2000, recognized them as such (Articles 5, 86, and 90). The current Law 4/2007 of 16 January, on Social Security Framework Law, envisages IPSS as collaborative entities within the Social Security system in the exercise of non-profit social action, and points to IPSS as entities that can receive public subsidies for the development of social action, included in the social care network with central government agencies, local authorities, and public institutions (Article 31). Likewise, Law 4/2007 reiterates the powers of supervision and inspection of the State over the IPSS (Article 32).

The Collaboration Protocols that regulated the IPSS collaboration procedure with the State that began in 1980 were continually renewed until, on 19 December 1996, the National Solidarity Cooperation Pact between the government, the Nation Association of Municipalities, the National Association of Parishes, the Union of Private Institutions of Social Solidarity, the Union of Mutualities, and the Union of the *Misericórdias* highlighted the special relevance of the latter among the different IPSS structures working in the social and solidarity sector, since, although the UIPSS was a signatory of this pact and the UMP was covered by this signature, the pact was also signed by the UMP.

Successive Social Security laws in Portugal have not always shared the same vision of social action in terms of the rights demanded by citizens or the welfare framework being dependent on the budget and available resources of the different institutions. Likewise, the weight that the IPSS carry in terms of social care, taking up to 75% of the public budget, has called into question whether they should be considered public bodies rather than private entities (Almeida 2010, pp. 157–58). However, all the laws that regulate Social Security in relation to IPSS have considered them to be private entities that are collaborators with the State in the area of public social care, all of which suggests that the *Misericórdias* are considered private social and welfare entities that collaborate with the State.

The role of the *Santas Casas de Misericórdia* in the provision of social and welfare services was formerly recognized in Law 30/2013 of 8 May, the General Framework of the Social Economy, in which Article 4 recognizes the *Misericordias* as a third type of entities that make up the social economy, listed after cooperatives and mutual societies, and before the foundations, the rest of IPSS, other entities covered by the community, and self-management subsectors integrated in the cooperative and social sector, as well as other legal structures that the government can subsequently incorporate into the database of the social economy that the law itself creates (Article 6), in which the *Casas Santas de Misericórdia* are legally recognized as one of the principal entities of the social economy in Portugal.

## 5. The *Santas Casas de Misericordia* in Spain

As mentioned in the introduction, the *Santas Casas de Misericórdia* (Holy Houses of Mercy) are entities that exist outside the strictly ecclesiastical scope that arose in Portugal in the late 15th century; instead, they are a lay structure for the care of the needy and for the fulfillment of all Catholic Works of Mercy. As also mentioned, the *Casas de Misericórdia* continued to expand during the 16th century throughout the territories of the Portuguese Empire.

In the neighboring empire of Spain, social care during the Middle Ages was essentially developed by ecclesiastical institutions, which assumed this responsibility almost exclusively, dedicating a good part of their income, convents, monasteries, and other institutions to the care of the needy, in the belief that poverty was something positive that set an example for the rich to renounce their wealth, and that the giving of alms was a proper means to achieve eternal salvation. The welfare institutions par excellence in the Middle Ages in Spain were the monastic hostels and the episcopal [10] xenodochiums [11]. These purely ecclesiastical establishments evolved from being chiefly religious sites where works of charity took place, to health centers associated with poverty and the care of the dying, in accordance with the demands of civil society and the responsibilities of the laity and municipal governments in matters of public order and citizen health. The management, which at first fell to the monasteries, and later to the bishops and military orders, by the 14th and 15th centuries ended up in the hands of the parishes, the religious brotherhoods, and the councils (Martínez García 2008, p. 81).

From the 16th to the 18th century, the discoveries of new lands, the rise of commodification, and the desire for expansion and improvement led to a change in how the poor, who were desacralized as an image of Christ, were regarded—ceasing to be an intrinsic part of the natural system of life and instead representing subjects for improvement, through a process that would make them useful citizens and thus contribute to the greatness of the State. Thereafter, the care of the destitute was reconsidered as purely a work of religious charity, introducing the idea of self-sufficiency—that is, those who can work and contribute cannot receive alms. This gave way to the ideas that work is an obligation for all and poverty is antisocial, dangerous, and reprehensible (De La Fuente Galán 2000, pp. 15–16), ideas close to Christian Reformism. A distinction was made between the legitimate or "worthy" poor whose personal and physical situation justified a degree of indigence, and the poor

---

[10]　The monastic attention to the needy in Spain was especially enhanced by the so-called Isodorian norm dictated by the Bishop of Seville Isidoro in 619, which obliged monasteries to dedicate one-third of their income to the support of the destitute (Alegre Peyrón 1984, p. 35), but this attention on the part of the monastic enclosures was so magnified that in the Council III of Zaragoza in 691, in its canon 3, the monasteries were exhorted not to become hospices for the laity (Díaz 2006, p. 20).

[11]　Free hospices for pilgrims and those needing medical care.

who sought alms as a means of living—their lack of employment was considered unjustifiable. Thus, care centers such as *Casas de Misericordia* were created to care for the worthy poor and, as far as possible, effect their transformation into productive citizens through the work and training provided. For the second type of poor, those deemed unworthy, orders prohibiting begging without specific permission were issued. In fact, during the time of Carlos I, in 1540, the so-called Tavera law, or law of the poor, stated that in each city there would be only one hospital for care of the needy and that begging on the streets was prohibited (Monge Jurado and López Osuna 1998, p. 33).

In Spain, after the Council of Trent in which the Church took over care and aid organizations (a situation that did not occur in Portugal), all care centers that were not established by royal imposition depended on the Church, even if they were administered by the laity (Villanueva 2009, p. 2). Thus, the development of hospital care became increasingly irrelevant for secular entities such as lay brotherhoods, or fraternal guilds, with the Church assuming the task that its lay brotherhoods in Portugal were carrying out. Likewise, following the Council of Trent, the imposition and control of religious observance became a focal point for the brotherhoods; in turn, the brotherhoods were jealously controlled by the parish priests, their immediate superiors in the Church hierarchy. By becoming integrated into the parish world and bringing with them all their acts of worship and charity, these secular entities were brought wholly under the control of the ecclesiastical authorities (Mantecón Movellán 1990, p. 1200).

Special mention must be made of the historical period between 1580 and 1640 regarding the role and ownership of the lay brotherhoods and the care services supplied by the *Casas de Misericordia*, at a time when Portugal and Spain were linked by the same kings of the House of the Habsburgs: Phillip II, Phillip III and Phillip IV of Spain (with an ordinal less, respectively, in Portugal). During this period, the development and actions of the Portuguese *Casas Santas de Misericordia* remained, like all other entities, institutions, and legal systems of the different kingdoms of the monarchy, under a polysynodial system of management by councils. The independence of these brotherhood entities from the Church was advocated by the Council of Trent and thus they remained under the royal control of the *Philippine* (from Felipe) dynasty.

Following the death in 1578 of King D. Sebastián I of Portugal, the grandson of Carlos I of Spain, and the immediate death of his successor, his grand-uncle, Enrique I, the so-called King-Cardinal, in 1580, who died without naming a successor, Phillip II, the son of Carlos I of Spain, asserted his dynastic legitimacy and annexed the Kingdom of Portugal. After taking possession of the Portuguese Crown, the new monarch retained the political–administrative structure of the Portuguese royal house (Labrador Arroyo 2006, pp. 24–31), without substantial modifications, although with a clear tendency towards bureaucratic control of the Portuguese institutions while still respecting their management (Cardim 2002, p. 32). This happened to such a degree that, in 1581, upon entering Lisbon as Philip II of Portugal, he became a member of the Brotherhood of *Misericórdia* of Lisbon (Abreu and Paiva 2006, p. 7).

The system of government used by the Habsburg Monarchs was known as polysynodial, with the king ruling in coordination with the synods or councils, as basic units of the Government (López Serrano 2001, p. 6). Some of the councils or synods Felipe II ruled had been inherited from his father, Carlos I, who used the Councils of Castile, Aragon, and Navarra for the territorial management of different kingdoms, with different laws, courts, and currencies; they were united only by the figure of a common king, and the councils of the new lands conquered in transatlantic territories, the Council of the Indies, and the Italian kingdoms, the Council of Italy, were added later. Along with these territorial councils, others of a more functional nature such as the Council of Military Orders, the Treasury Council, the Inquisition Council, and the State Council were also in operation. For the management of the kingdom of Portugal, Felipe II created the Council of Portugal (Escudero 2003, pp. 18–19), through which the Portuguese kingdom retained its political and administrative structures with some degree of autonomy and full recognition within the Hispanic Monarchy, in addition to having the express support of the Portuguese Church, which helped it to maintain its privileged status quo compared to its Castilian counterpart (Palomo 2004, p. 64). This recognition and treatment meant that the development of the Portuguese *Misericórdias* was not affected by the

dynastic change from the House of Aviz to the House of Austria. However, in the rest of the Hispanic kingdoms, the same brotherhood model was not reproduced.

With particular regard to Spain, in the modern era the term *Casa de Misericordia* did not refer to the same brotherhoods that emerged and developed in neighboring Portugal. At the end of the 18th century, in the development of the care policy in Spain, the Council of Castile, when recognizing the need for the creation of spaces destined for the care of vulnerable groups, the *Casas de Misericordia* were understood to be "all those destined for lodging, or common asylum of some class of poor, who due to their short life, or old age, or by any other circumstance they are invalid, or it suits the public cause that they live united for some time. Inmates of orphaned children, the homeless, elderly, disabled, widows, and any miserable people, are deemed to be a Casa de Misericordia. These also include the Hospitals of the Sick, Foundling Homes, and the Galeras (women´s prisons), and Correction Houses for men, and poor women ... under the title of Casa de Misericordia is understood to be any shelter, that is destined for refuge of some kind of needy" (De Murcia y Córdoba 1798; pp. 2, 10). The concept of the *Casa de Misericordia* that existed at the end of the 18th century in Spain dated back to the era of Greek splendor and the so-called hospitality houses for the relief of the poor, which were called "Hospitals of Jupiter," as well as those public houses of *Misericordia* that were created by the Church in the 8th century after the reign of Constantine I (of De Murcia y Córdoba 1798, pp. 4–5), in such a way that any hospitality houses—hospitals—of various kinds (for the poor, orphans, or the elderly) are considered *Casas de Misericordia*.

As part of the social policy that developed during the Ancien Régime in Spain, hospices were the care establishments most often called *Casas de Misericordia* (from De Murcia y Córdoba 1798, p. 11). These hospices were not establishments for foster care, the maintenance and education of the poor, foundlings, and orphaned minors as is currently understood (RAE), but rather were public houses where the underprivileged poor were gathered for care and sustenance and where training and work was sought for those that were able-bodied. Already in the time of Philip II, in 1597, by royal order, these types of shelters existed in 50 cities and villages for the accommodation of disabled poor people, with the aim that only those who had been duly identified and had a "written license of Justice" would be permitted to beg (from De Murcia y Córdoba 1798, pp. 38–39).

As in the rest of Europe, the secular brotherhoods that developed in the Hispanic kingdoms during the Middle Ages, in addition to mutualistic guilds, devotional and religious organizations of the feudal system, came to fulfill the need of helping the poor and destitute, but they differed according to the type of care—some were dedicated to caring for the sick, others to the disabled, the elderly, or the unemployed; some accommodated orphans and widows; and others were devoted to various type of charity work such as providing dowries for poor maidens or relief to prisoners. All these charitable organizations operated without general oversight, as occurred with the Portuguese *Misericórdias*. During the 16th century, the work of the brotherhoods was related, fundamentally, to helping the sick, either in their own centers or in public hospitals. This work carried out by the brotherhoods was little more than the fragmentation of acts of charity, as opposed to the more centralized and secularized initiatives of social care promoted by the Crown (Mantecón Movellán 1990, p. 1202). During the reign of Felipe II and his project for reducing the number of hospitals in order to centralize and optimize resources, a great many of these brotherhoods disappeared; those that remained were, fundamentally, for the spiritual care of accompanying the sick, and if necessary, for economic attention to the deceased's debts and burial expenses (Lozano Ruiz and Torremocha Hernández 2013, pp. 23–26).

Hospital care during the 16th century in the Courts of Castilla focused on the creation of singular general hospitals. The Courts of Toledo in 1525 and Segovia in 1531 promoted the creation of these hospital facilities, within a general framework of regulating almsgiving and care policies in addition to repressing idleness, begging, and vagabonds. Thus, with the Order of 24 August 1540, which regulated hospital care (Valenzuela Candelario 2007, pp. 338–39), the brotherhoods and private hospitals had to step back so as not to divert funding in the form of alms, or testamentary provisions, to small institutions instead of to central hospitals. This meant that the hospital centers were deprived of private management and ownership.

Outside the hospital framework of the brotherhoods, by the end of the 16th century, a large number of *Casas de Misericordia* were created to solve the problem of the many poor and vagabonds of both sexes and of all ages that roamed the cities of the time. They also provided care services for imprisoned poor women, beggars, orphans, the homeless, vagrants and/or petty criminals[12]. These *Casas de Misericordia* were referred to as "Correction Houses," and although they were mostly destined for women, they were not exclusively so, also being used for the reeducation of the so-called "intractable and naughty boys" (from De Murcia y Córdoba 1798, p. 92). From the 18th century onwards, the creation of *Casas de Misericordia* for the housing of abandoned children, the so-called *Casas de Expósitos* (Foundling Homes), represented a new educational approach under King Carlos III as well as a new financing system, the so-called *Fondo Pío Beneficial*[13] (Sánchez Vázquez and Guijarro Granados 2002, pp. 123–24), which continued during the reign of Carlos IV. The other care facilities that were included within the so-called *Casas de Misericordia* were the "sick hospitals," especially the so-called "*lazarinos*" (leper houses), which were, with the exception of Granada, scarce and often poorly managed. All these health care spaces were known as *Casas de Misericordia* in a generic sense to denote spaces of care and assistance. Their ownership, in the vast majority of cases, was ecclesiastical in nature, by dioceses, or sponsored by the monarchy. In terms of funding, this was also from ecclesiastical, or public funds, be they royal or municipal. As a result, the *Casas de Misericordia* that were actually owned by the brotherhood or other lay fraternities were so scare that their existence was considered merely symbolic, a fact that is acknowledged in Royal Decree of 11 December 1796, which contains the Regulations for the Control of Abandoned Children.

With the discovery of America and the expansion of the Empire, Spanish society changed dramatically. People without resources could be soldiers; however, on completing their military service, many were unable to find civilian employment and were unable to access social care. Likewise, the influx of riches from the conquered territories discouraged the State and its subjects from developing other industries for their livelihood. Given this scenario, work was devalued and an "idle" way of life became common, as did picaresque behavior, with the appearance of the "false poor" who faked or inflicted physical losses so as to live on charity without working. This problem began during the reign of Phillip II and continued until the end of the 18th century, when Enlightened Despotism, personified by Carlos III and Carlos IV, sought a solution to this situation by empowering the *Casas de Misericordia* to care for the "true poor." The architectural structure of these *Casas* was also divided into seven departments: two of these were segregated work areas for men and women who were not confined in the *Casa*; the third and fourth departments were for orphaned girls and boys; the fifth and sixth were for men and women who were deemed to be lazy and vicious and needed to be reformed; and lastly, the seventh department was for prostitutes and abandoned women who worked in the *Casa* as part of their sentence set by the Courts (De Murcia y Córdoba 1798, pp. 127–31).

After the fall of the Ancien Régime and the arrival of the Liberal State, triumphant liberalism in the first half of the nineteenth century led to the displacement of the charitable works of the Church (whose assets were confiscated and their work separated from public works) or the Community with regards to the care of the marginalized and the impoverished. This also, following the Decrees of 8 June 1813 and the Royal Decree of 20 January 1834 that prohibited trade ordinances, led to the disappearance of the guilds as a means of worker protection. The work of these organizations was, in large part, replaced in Spain by Friendly Societies, charitable–welfare institutions, and the *Montes de Piedad* (a kind of pawnshop), along with existing ecclesiastical charities and other non-profit

---

[12]　In fact, until the 19th century, the *Casas de Misericordia* represented the most important institution for the seclusion and assistance of these women considered "deviant" (Almeda Samaranch 2005, p. 79), but criminals needed to enter the female prisons, called Casas de Galera, in order to fulfill their sentences.

[13]　The Pius Benefit Fund was granted by Pope Pius VI to the Spanish monarch, Carlos III, on 14 May 1780, to withhold one-third of the value of certain income from the privileges and ecclesiastical benefits for the support of the hospices, the relief of the begging, and the useful employment of the poor; his successor Carlos IV ordered a reduction to a tenth, which would be administered by the bishops (Canga Argüelles 1833, p. 470).

organizations, went on to establish and institutionalize State-owned charities funded by the resources and institutions of the Church (through the confiscation of their assets). Given that poverty had become a matter of public order and police action was needed to control the homeless and unlicensed beggars, the process of rationalization and centralization of health care facilities initiated by the Enlightened Monarchy was accelerated (Fernández Riquelme 2007, pp. 23–27).

It was at this time that the social welfare system was institutionalized, which reduced dependence on the charity of the religious and private sectors. With the Law of Primogeniture and Entailed Estates of 12 October 1820, which was an antecedent of the confiscation laws, all ecclesiastical and foundational institutions with real estate were abolished. The Decree of 27 December 1821 established public hospitals in all the provincial capitals and towns in which the Government deemed it convenient (Article 105). Likewise, under the successive Public Charity Acts of 1822, 1836, and 1849, social care was brought under public control, to wit, Article 1 of the Law of 1849 stated, "charitable establishments are public." Despite this decree, some private charities were allowed to continue, although they were placed under public control and supervision, distinguishing them from provincial and municipal charities (Article 2). Likewise, the management of all charities, irrespective of the type of services offered or their ownership structure, was placed under government control (Article 4), whether under national, provincial, or local administration, depending on the ownership or scope of action. The different types of social care establishments were categorized into maternity homes, charitable homes, charitable asylums, and hospitals, in addition to other categories such as municipal relief funds and public charitable funds. A Charity Board was established to manage all the care establishments and allocate economic resources, in addition to a more general Advisory Board to work alongside the relevant public administration, whether provincial or municipal. Thus, the welfare policy of the kingdom was deemed to be a public obligation, and private establishments were placed under a monitoring and intervention regime that eventually led to their disappearance.

The nineteenth century was a time when different social typologies of a private nature arose for the care of injuries and suffering caused by work. Mutual aid societies arose, recognized by the Royal Order of 28 February 1839, and are defined as "the corporations constituted to render mutual aid in times of misfortune, illnesses, etc., and pool economic resources in order to attend to future needs." These protection entities were of a professional character, filling the void left behind by the trade guilds, and are considered to be the embryos of future trade unions, although, in practice, they scarcely warranted any kind of relevance in terms of meeting the challenge of accidents or illnesses (López Castellano 2003, p. 2008). Social care institutions already depended on public entities, be they local, provincial, or national, as well as ecclesiastical authorities or even a combination of the two. This meant that, while some specialized traditional social care facilities such as *Casas de Misericordia* remained, most of them disappeared as their work was increasingly assumed by public entities.

By the end of the 19th century, in light of what was occurring in other European countries, the protection of workers became institutionalized with the creation in 1883 of the Commission of Social Reforms as a result of workers' struggles for the right to association and social protection and welfare. This was followed by the creation of public bodies such as the Institute of Social Reforms in 1903 and the National Institute of Social Security in 1908 (Juárez and Sánchez Daza 2004, pp. 201–2). Since the end of the 19th century and still in the present, social care in Spain has been characterized by public ownership and responsibility, although sharing the obligation of care in the case of eventualities arising from work-related accidents, which may last until the retirement of the worker. In this scenario, private entities have a merely token presence in social care outside of the framework of workers´ compensation or the private hospital network, be they funded by private hospitals, whether in turn these were founded or sponsored by mutual insurance companies, private universities, or purely capitalist business entities, as well as those owned by the Catholic Church. That is to say, private entities that provide social care services are not specially empowered by public assistance policies.

There are currently very few *Casas de Misericordia* in Spain that provide social care services. These generally have retained their original legal structures, some lay and some ecclesiastical, and many of

them are also enmeshed with either local or provincial public administrations. As a sample, the following can be considered to be the most relevant:

-   The *Santa Casa de Misericordia of Pamplona*, which is focused on the care of the elderly, was founded in 1706, and currently has the legal form of a foundation with a mixed Board of Trustees composed of members of the City Council of Pamplona who appoint the President and four members, and with a high number of neighborhood ordinary members. The *Casa*, in addition to its social function, has been, since 1921, the owner of the bullring and organizer of all bullfights that are held in Pamplona, as well as the running of the bulls during the Festival of San Fermín[14].
-   The *Casa de Misericordia of Barcelona*, founded in 1581 as an asylum for the homeless poor, has been, since 1984, a private foundation, administered by a majority of Board members from civil society, an employer representative of the Archbishopric of Barcelona, and another representative of the Prior of the Holy House. This *Misericordia* is focused on the care and protection of children at risk of social exclusion, managing a children's residence, a house and camps in the countryside, and a residence for students that grants scholarships for those without financial resources.
-   The *Provincial Hospital of La Misericordia of Toledo*, which began under the management of the Brothers of Mercy in the late 15th century, and, after the Law of Public Charity of 1822, had its management granted to the Daughters of Charity of Saint Vincent de Paúl, in 1847, was declared to be a municipal establishment dependent on the Municipal Charity Board. In 1859, as with most hospitals owned by the *Misericordia,* it came under public ownership and became a provincial hospital (Gómez Rodríguez 1991, pp. 102–22).
-   The *Santa Casa de Misericordia of Bilbao*, founded in 1724 as a Refuge for Outsiders, is currently run as a nursing home and part of the Lares Euskadi Association, and is managed by the Daughters of Charity of Saint Vincent de Paúl, as part of the Charity Board of both the City Council of Bilbao and the Provincial Council of Bizkaia. Its legal structure is that of a private charity established in accordance with Article 38 of the Civil Code, established under the current civil legal regime, by autonomous legislative powers, in accordance with Law 9/2016 of 2 June, on the Foundations of the Basque Country, although legally structured as a foundation rather than a private charity, as set forth in Decree 359/1985 of 12 November, on the creation of the Registry of Foundations and Charitable-Aid Associations and Similar Entities of the Basque Country, where the *Santa y Real Casa de Misericordia de Bilbao* is registered.
-   The *Santa Casa de Misericordia of Olivenza* is possibly the only one whose activity can be identified with the *Misericórdias* in Portugal—unsurprising given that, until 1801, it was deemed to be part of Portugal. The so-called "Question of Olivenza" meant that Portugal refused to recognize the sovereignty of Spain over the city (a similar situation to the case of Táliga, also in the province of Badajoz, and which turned out to be the most "Portuguese" of Spanish populations). The *Misericordia* of Olivenza, founded in 1501, currently has the legal structure of a private foundation, and so is subject to the national legislation for foundations. Its organizational chart is similar to that of the Portuguese entities, with the existence of a *Provedor* (President), an *Escribano* (Notary Public), and a treasurer, and, as occurs in the Portuguese *Misericórdias*, it provides a comprehensive range of general social care for those in need, beyond the elderly and the sick. It includes a residential center for the elderly and disabled, a training center for the unemployed, and in-home care services for dependent adults and the elderly.

## 6. The *Misericordias* as Social Economy Entities in Spain

The Spanish Constitution of 1978 expressly includes in its preamble democratic coexistence within the Constitution and "laws in accordance with a fair economic and social order," and proclaims in Article 1.1 that "Spain constitutes a social and democratic State, subject to the rule of law. In Article 38, it is determined that "Freedom of enterprise is recognized within the framework

---

14   http://feriadeltoro.com/plaza-de-toros-de-pamplona/historia/.

of the market economy ... in accordance with the demands of the general economy and, as the case may be, of economic planning."

At the time of the Constitution, there was no consensus in Spain on what should be understood by the social economy. However, the Constitution already contained, on the one hand, a socialized concept of economic activity, as foreseen in Article 33.2, where the right to private property and inheritance is recognized: "the social function of these rights shall determine the limits of their content" and (Article 128) "The entire wealth of the country in its different forms, irrespective of ownership, shall be subordinated to the general interest." On the other hand, there are also specific references in the Constitution to entities that are integrated within the specific sector of the social economy. Thus, Article 129.2 points out the importance of cooperative societies, stating that "The public authorities shall efficiently promote the various forms of participation in the enterprise and shall encourage cooperative societies by means of appropriate legislation. They shall also establish means to facilitate access by workers to ownership of the means of production." Article 22 also recognizes the right of association and Article 34 enshrines the right to create foundations: "The right to set up foundations for purposes of general interest is recognized for purposes of general interest."

Therefore, although the term social economy is not manifestly present in the text, the Constitution does deal with those organizations that have traditionally been part of the social economy, with even a certain sector of law writers considering that Article 41 of the Spanish Constitution, which determines the obligation of the public authorities to maintain a public Social Security regime while recognizing the freedom of assistance and complementary benefits, is in fact admitting and promoting the activity of the Union of Mutualities, a mainstay of the social economy sector (Barea Tejerio 2003, pp. 137–38).

Aside from the Constitution, in the institutional context of Spain, the explicit acceptance of the social economy is manifested in the creation of *Instituto Nacional de Fomento de la Economía Social* (National Institute for the Promotion of the Social Economy) by means of Law 31/1990 of 27 December, of the General State Budgets for 1991, wherein Article 98.1 states that: "reporting to the Ministry of Labor and Social Security, the National Institute for the Promotion of Social Economy is created as an administrative autonomous body," with the third clause of this article designating, among its roles, "the promotion and development of Cooperatives, Labor Corporations, Labor Foundations and any other entities that in the future are determined by Law, coordination with the Ministerial Departments that carry out promotion actions in the scope of the Entities mentioned above."

Royal Decree 1836/1991 of 28 December 1991, which determines the basic organic structure and functions of the National Institute for the Promotion of Social Economy, as the managing body for the development of social economy policy, identifies in Article 2 the conditions that the so-called social economy entities must meet, specifically mentioning Worker-owned Societies. The Institute finally disappeared as an autonomous body, and its powers were first assumed by the General Directorate of Social Economy and, later, by the "General Directorate of Social Economy, Autonomous Labor and Social Responsibility of Companies," under the auspices of the Ministry of Labor.

It should be noted that it was the National Development Institute that promoted the creation of institutions of vital importance in the sector, such as the *Confederación Empresarial Española de la Economía Social* (CEPES) (Business Confederation of the Social Economy in Spain).

However, while the absence of specific regulation of the sector at a national level persisted, at the autonomous regional level, there was a repository of relevant legislation on the social economy in the respective Statutes of Autonomy.

Given this background, Spain undertook the task of preparing a national law that would specifically address the normalization of the social economy and that would also make it more visible. Through the Council for the Development of the Social Economy, and with the agreement of CEPES, the government commissioned CIRIEC (International. International Centre of Research and Information on the Public, Social and Cooperative Economy) in Spain to gather a group of experts to form a commission that produced a report for the purpose of drafting a Social Economy Law by

October 2009. On 30 December 2010, the Social Economy Law Project was published in the Official Gazette of the General Courts. The project, although with certain modifications and reforms carried out at the parliamentary headquarters, became enshrined in the current Law 5/2011 of 29 March, on the Social Economy, at the national level[15]. This was the first substantive regulation by a country of the European Union that espoused generating wealth based on the criteria of equity, democracy, and a focus on people and the environment. The Preamble of the Spanish Law emphasizes that the social economic sector is endowed with a high degree of legal certainty. Likewise, it sets guiding principles for all entities deemed to be part of the sector and a list of those entities that qualify as players in the social economy, leaving open the possibility of expanding the number of entities by adding others that meet the guiding principles set forth in Article 4 of the Law, and incorporating the government into the catalog of social economy entities (Article 6).

Among the social economy entities listed in Law 5/2011, Article 5, in addition to cooperatives or mutual societies, includes foundations and associations that carry out economic activities. While entities such as the *Misericordias* are not specifically listed, as is the case in Portuguese law, the fact that foundations are mentioned as entities of the social economy means that social care entities such as the *Casas de Misericordia* in Spain that are legally structured as private foundations or associations also qualify as social economy entities. It should be noted that it is neither the fact that the *Misericordias* were initially founded as charities, nor their historical trajectory as providers of social care, that qualifies them as social economy entities, but rather the adoption of one of the legal structures provided in Law 5/2011. Examples of this type of private foundation are the *Misericordias* of Olivenza, Pamplona, and Barcelona. Other *Misericordias*, in fact the majority, became public institutions in the mid-19th century, and, as such, do not qualify as entities of the social economy as their activity is not carried out in the private sector (Article 2 Law 5/2011).

## 7. Conclusions

*Las Misericordias* are secular brotherhoods or fraternities founded in Portugal at the end of the 15th century, starting with the *Casa Santa de Misericórdia* (Holy House of Mercy) in Lisbon, which sought to carry out the 14 works of Christian mercy. The *Misericórdias* spread throughout Portugal, as well as to the newly discovered colonies and territories of the greater Portuguese Empire.

In the sixteenth century, the *Misericórdias* became the principal institution for social care across the Portuguese Empire. At the present time, they collaborate with the State for the provision of Social Security services but continue to be a private entity, and are included in the so-called Particular Institutions of Social Solidarity, having constitutional recognition, being categorized as a social economy entity, benefitting from public policies for development of the social economy, and enjoying the tax advantages granted in Law 30/2013 for the Framework of the Social Economy. Thus, it can be said that the Portuguese *Misericordias* enjoy the status of social economy entities.

The *Misericordias* in Spain, founded some years later, were based on the Portuguese *Casas*. However, they have always been linked to a specific charitable act, unlike the more generic charitable works carried out by their Portuguese counterparts. The Council of Trent in the mid-16th century, which set the guidelines for the ecclesiastical control of all welfare brotherhoods, as well as establishing public policies for social care in an effort to counteract enlightened despotism and the Liberal State, made it very difficult to maintain private ownership of social care facilities.

In Spain, since the publication of the different charity laws, almost all the *Casas Santas* that continued to provide social services were taken over by local and provincial administrations, to the point that their social presence since the end of the 19th century has become purely symbolic.

In Spanish Law 5/2011 on the Social Economy, among the possible entities that make up the sector and participate in different programs and public policies for the development and future

---

[15]　At the autonomous level, based on the provisions of Article 30.1 of Organic Law 1/1981 of 6 April, on the Statute of Autonomy of Galicia, as a development of its competence in promotion and planning of the economic activity of Galicia, Law 6/2016 of 4 May on the social economy of this autonomous community was published.

protection of their entities and of the anticipated benefits of Social Security, there is no mention of the *Casas de Misericordia*. This notwithstanding, some of the surviving entities have adopted legal structures that are included in the Law, such as foundations. The Spanish *Misericordias* do not, per se, enjoy the status of a Social Economy entity. Only those that have adopted a legal form in accordance with the Law are considered entities of the social economy.

The Portuguese and Spanish *Misericórdias* thus have the same origin, but different institutional and legal trajectories.

**Author Contributions:** A.J.M.R. Investigation; Writing—Original Draft; Writing—Review and Editing; J.R.P.M. Investigation; Writing—Original Draft; Writing—Review and Editing; M.E.M.R. Investigation; Writing—Original Draft; Writing—Review and Editing; J.d.P.V. Investigation; Writing—Original Draft; Writing – Review and Editing. All authors have read and agreed to the published version of the manuscript.

**Acknowledgment:** This article was written during a research stay at the University of Beira Interior (Portugal). We are grateful to this University for the support and facilities provided during this period. Work carried out in the framework of the Centre for Research in Social Economy Law and Cooperative Enterprise of the University of Almeria. «NECE-UBI, R&D unit funded by the FCT—Portuguese Foundation for the Development of Science and Technology, Ministry of Education and Science, University of Beira Interior, Management and Economics Department, Estrada do Sineiro, 6200-209 Covilhã, Portugal».

**Funding:** This research received no external funding.

**Conflicts of Interest**: The authors declare no conflict of interest.

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
