# Peer review of "The Misericórdias as Social Economy Entities in Portugal and Spain"

_religions, doi:10.3390/rel11040200_

Round 1

Reviewer 1 Report

This paper has an excellent amount of history and does show promise to be a great history paper. The authors clearly spent a large deal of time researching historical records to compile this manuscript comparing and contrasting the Misericordia in Spain and Portugal. However, there are some serious concerns that need to be addressed within the manuscript, particularly when it comes to organization.

The introduction to this paper is not an introduction. It is a well done history section consisting up nearly half the paper, albeit some of the paragraphs that are not focused on the Misericordia could be reduced to keep the reader's focus on the topic at hand. Additionally, the history could use some organization with subheadings in order to better communicate what the author(s) most want to convey out of the given topic. Changing this section to a a history section and adding an actual introduction to introduce the Misericordia, their importance as a charitable organization, and describing what the main purpose of this research is would help this paper tremendously. 

The lack of an introduction compounds the main problem within the research: the purpose of the research is never directly stated. I was not aware that the purpose and contributions of this paper consisted of a comparison and contrast of the Portuguese and Spanish Misericordias until the final sentence in the paper. It is implied in the abstract, but this point absolutely needs to be made clear in the introduction.

Finally, the paper could use a rewrite consolidating many of the different ideas. Possibly showing first how the Spanish and Portuguese Misericordias began similarly, what institutional forces required them to divulge, and how they are important both as religious and as non-profit organizations today. The authors have done a pretty good job of insuring that most of this content is in the paper, it just needs to be written in a clear, concise way to fully understand their contributions.  

Author Response

First of all, I would like to thank the reviewer for his accurate comments on the work presented. They have undoubtedly served to substantially improve its quality and gain academic precision.

In relation to his comments, we have added an introduction pointing out the objectives that we intend to pursue in the development of the work, with the research method and anticipating the fundamental causes that caused the differentiation of the Portuguese and Spanish mercies.

Likewise, we have added a new epigraph to the work to lighten the one we titled "The Holy Houses of Mercy in Portugal (Origin and expansion)", using part of the content of that epigraph as an introduction to the origin of the attention to the needy and which is common to both countries.

We have also highlighted the main causes of the differentiation in the development of the Portuguese and Spanish mercies, as well as increased the footnotes to avoid distraction in the development of the main exhibition.

Thank you for your contributions.

Reviewer 2 Report

This is an excellent article. 

  1. One quick edit -- do not confuse Spainish with Portuguese! That is, it is not "Las Santas Casas de Misericordias" in Portugal, it is "As Santas Casas de Misericordias." 
  2. I think that the main point of the article, "The Portuguese and Spanish Misericordias: the same origin, but different institutional and
    legal trajectory" is a brilliant insight, and a correct statement
  3. The history of the Misericordia in Iberia is excellent.
  4. The overall tone of the article is fair and academic.
  5. I would like to see a bit more appreciation of the deep faith of Queen Leonor of Portugal. Arguably, her faith and devotion to the Gospel was where the original notion of the Misericordia came from in 1498 (the same year that Vasco da Gama reached India).
  6. Queen Leonor was not driven in a cynical way to expand state power; she wanted to place state power under the clear dictates of the Gospel. 
  7. So, I think that the original vision needs to be more clearly stated. It was a beautiful effort of combining faith and action.
  8. The article is right to note that the Misericordia remains an essential part of Portuguese social services today. Some have argued that the entire welfare state would collapse in Portugal without it. 
  9. The case of Spain is not as clear, is derivitive of the Portuguese experience, but still is an effort to combine faith and good works.
  10. Spain did not have the great Queen Leonor, so the Misercordia there looks different. However, this observation is correct:"In Spain, the existence and survival of the Santas Casas de Misericordia does not have the same long history, nor the same social relevance as their Portuguese counterparts. However, even today, there are some Casas de Misericordia in Spain that provide social care services, having adopted various legal structures such as foundations associations and public entities."
  11. The author is quite right to point out the similiaries and the differences between Spain and Portugal: " Their trajectory has not been linear, but rather they have suffered many upheavals, relevant to their time, but have always played an important role in the social care of citizens."
  12. It would be interesting to place a footnote observing that the Portuguese built an impressive network of Misericordias throughout the colonies; in Brazil, Angola, Mozambique, and elsewhere. Leonor's vision is still at work.
  13. Even Portuguese immigrants to France have launched a "Santa Casa da Misericórdia de Paris." It's such a powerful idea and statement of appllied faith. 
  14. Congratulations on a wonderful article on an important subject.

Author Response

First of all, we would like to thank you for your words of congratulation, as well as for your accurate comments on the work presented. They have undoubtedly served to substantially improve its quality and gain academic precision.

We have tried to reinforce the vocational spirit of Queen D. Leonor with the incorporation of new tapes and three research articles in the bibliographical references.

With regard to the comment that "Queen Leonor was not driven in a cynical way to expand state power", we have tried to reinforce the queen's evangelizing idea, although there are authors like the one we cite who add to the spiritual component for the development of the mercies, a political interest in bringing the monarchy closer to the population.

Finally, we have added an introductory epigraph, emphasizing the expansion of the Portuguese mercies to other territories in the area of influence of Portugal, as well as the new ones that have been created as a result of the emigration of Portuguese nationals, such as those from Paris and Luxembourg.

Thank you for your contributions.

Reviewer 3 Report

This is a well-researched history of the origin and long existence of the Misericordias in Portugal and Spain and has an interesting puzzle embedded in it, namely why they continue to play an important role in social provision in Portugal but do not in Spain.

My recommendation would be to restructure the paper along the following lines: presentation of the puzzle and its significance, brief description of the origins and activities of the Misericordias over time, answering the puzzle, what the difference tells us about state development in Portugal and Spain. 

The article as currently structured presents a very long chronology and history of the Misericordias but the reader is unsure about the point of this descriptive information and the overall argument until the very short conclusion, and even then is unable to determine the significance of these differences.  Does Portugal do a better job serving the vulnerable than Spain? Does it matter that Misericordias are weaker in Spain, if the public sector or other third sector organizations take care of the vulnerable?  What larger lessons about the development of the state, or the church, or the social sector can be drawn from these differences?

Author Response

First of all, we would like to thank the reviewer for his words and his accurate comments on the work presented. They have undoubtedly served to substantially improve its quality and gain academic precision.

We have restructured the document by adding an introduction pointing out the objectives we intend to pursue in the development of the work, with the research method and anticipating the fundamental causes that caused the differentiation of the Portuguese and Spanish mercies.

We have also incorporated a new section to the work to lighten the one we used to call "The Holy Houses of Mercy in Portugal (Origin and expansion)", using part of the content of that section as an introduction to the origin of the care of the needy and which is common to both countries.

We have also highlighted the main causes of the differentiation in the development of the Portuguese and Spanish mercies, as well as increased the footnotes to avoid distraction in the development of the main exhibition.

We have tried to highlight the causes that provoke the different institutional and legal treatment of the mercies in the two countries, and the conceptions of social care that have been developed in Portugal and Spain, with their derivation of recognition and incorporation of the social work that the mercies carry out in Portugal, and the difficult situation of survival that the Spanish mercies have due to the assumption of the public obligation of social care, without prejudice to recognizing other private formulas of care for the vulnerable, which has been highlighted in the conclusions.

Thank you for your contributions

Reviewer 4 Report

General comment
The article is interesting and contains originality with regard to the comparison that the author makes, in a comparative perspective, of the historical development of "Misericordias" in Portugal and Spain, ending up addressing a relevant contemporary theme such as the social economy.
However, there are aspects that could be improved:
1) Clear definition of the purpose of the article and the methodology and type of sources adopted.
2) Reflection, even if brief, of the process of secularization of the two societies under study, using the transformations of "Misericordias" as a lens for understanding the broader macro-societal transformations.3) A greater detail regarding the political changes in Spain during the 20th century, similar to what the author does in relation to Portugal.

4) Insert a note on the current perspective of social economy.

There should also be a careful review of the bibliography and references in the text. For example:

In page 4, lines 155-156 should be "Sá" and not "Dos Guimarães" and in Bibliogrphy it should be "Sá, I. dos Guimarães" instead of "Dos Guimaraes Sá, I." or "Dos Guimaraes Sá" 

The author should also take in account the publication below. Although focused on the Misericordia in Porto (Portugal), it is a recent work of unavoidable relevance.

Amorim, Inês (2018) - Sob o manto da Misericórdia - Contributos para a História da Santa Casa da Misericórdia do Porto. Porto: Almedina (4 volumes. Volume I - 1499-1668; volume II – 1668-1820; volume III – 1820-1910; volume IV - 1910 aos nossos dias).

Author Response

First of all, we would like to thank the reviewer for his words and his accurate comments on the work presented. They have undoubtedly served to substantially improve its quality and gain academic precision.

We have restructured the document by adding an introduction pointing out the objectives we intend to pursue in the development of the work, with the research method and anticipating the fundamental causes that caused the differentiation of the Portuguese and Spanish mercies.

We have also incorporated a new section to the work to lighten the one we used to call "The Holy Houses of Mercy in Portugal (Origin and expansion)", using part of the content of that section as an introduction to the origin of the care of the needy and which is common to both countries.

With regard to the process of secularization referred to in his commentary, it should be noted that from their origin the mercies had a secular character and that following the Council of Trent, in Spain, as in the rest of the Catholic world, with the exception of Portugal, the ecclesiastical character of this type of confraternity was imposed. Portugal has continued with the secular character of the mercies until practically the 20th century in which the Church and the State, but not the mercies themselves, have conceptualized these confraternities as belonging to the canonical sphere, but we have not entered into the conflict that arose around the nature and legal regime to be applied to the Mercies, which has been partially resolved by the Catholic jurisdictional entities but which is still latent, as it is understood that, for the time being, it does not affect the nature of the social service provided by the mercies and the legal recognition of them by the Portuguese legislation as IPSS, a collaborator of the Social Security system and an entity of the Social Economy.

On the other hand, we have not developed further the political changes that have taken place in Spain in the twentieth century because, in reality, the legal and institutional significance of the Misericordias in Spain reached the nineteenth century with the various welfare laws. The 20th century in Spain has not changed, at least significantly, the legal and institutional situation of the Casas de Misericordia, except for the publication of the 2011 Social Economy Law, insofar as it may affect certain mercies that have adopted one of the structures envisaged for such entities, which is the subject of this paper.

We have corrected the citations in the text, and incorporated the recommended bibliography.

Thank you for your contributions.

Round 2

Reviewer 1 Report

The authors took the comments from myself and my fellow reviewers seriously and it clearly shows in their new version. This draft is much more coherent and highlights the excellent history research the authors have put forth more clearly. However, there is one slight modification that I would like to see to really make this paper the best that it can be. The introduction is much better; however, it still lacks a clear statement of the research question. This can easily be added at the end of the paragraph from line 27-32 starting with "However." Outside of that, this paper's content remains excellent and it's organization has improved dramatically. 

Author Response

Thank you very much for your comments, we really appreciate them. We have included your suggestion at the end of the introduction (lines 84-88)

Reviewer 3 Report

Thank you for your responses as well as to the revisions to the paper, particularly with respect to the introduction. These have strengthened it considerably by making the argument clearer. Well done!

Author Response

Thank you very much for your comments, we really appreciate it.